An examination of disparities in cancer incidence in Texas using Bayesian random coefficient models

Sparks Corey corey.sparks@utsa.edu
Department of Demography, The University of Texas at San Antonio , San Antonio, TX , USA
Department of Biostatistics and Epidemiology, University of Texas Health Science Center at San Antonio , San Antonio, TX , USA
Evans D. Gareth
Electronic publication date: 2015 Sep 24
Publication date: 2015
Volume: 3
Electronic Location ID: e1283
Received 2015 Jan 29; Accepted 2015 Sep 9
Copyright: © 2015 Sparks
Copyright year: 2015
Copyright holder: Sparks
License: This is an open access article distributed under the terms of the Creative Commons Attribution License, which permits unrestricted use, distribution, reproduction and adaptation in any medium and for any purpose provided that it is properly attributed. For attribution, the original author(s), title, publication source (PeerJ) and either DOI or URL of the article must be cited.
License URL: https://creativecommons.org/licenses/by/4.0/

Keywords: Health disparities, Bayesian modeling, Cancer incidence, INLA

Funding: Cancer Prevention Research Institute of Texas RP120462 Partial funding for this work was provided by the Cancer Prevention Research Institute of Texas Award number RP120462. The funders had no role in study design, data collection and analysis, decision to publish, or preparation of the manuscript.

==============================
Disparities in cancer risk exist between ethnic groups in the United States. These disparities often result from differential access to healthcare, differences in socioeconomic status and differential exposure to carcinogens. This study uses cancer incidence data from the population based Texas Cancer Registry to investigate the disparities in digestive and respiratory cancers from 2000 to 2008. A Bayesian hierarchical regression approach is used. All models are fit using the INLA method of Bayesian model estimation. Specifically, a spatially varying coefficient model of the disparity between Hispanic and Non-Hispanic incidence is used. Results suggest that a spatio-temporal heterogeneity model best accounts for the observed Hispanic disparity in cancer risk. Overall, there is a significant disadvantage for the Hispanic population of Texas with respect to both of these cancers, and this disparity varies significantly over space. The greatest disparities between Hispanics and Non-Hispanics in digestive and respiratory cancers occur in eastern Texas, with patterns emerging as early as 2000 and continuing until 2008.

Introduction

Disparities in cancer incidence and mortality exist between racial and ethnic groups in the United States and worldwide (Du et al., 2007; Elmore et al., 2005; Harper et al., 2009; Hun et al., 2009; McKenzie, Ellison-Loschmann & Jeffreys, 2010; Siegel, Naishadham & Jemal, 2012; Vainshtein, 2008). The causes of these disparities have been suggested to be rooted in different levels of socioeconomic status (SES), access to medical care, differential exposure to carcinogenic materials and differential treatment by medical staff of racial and ethnic minorities (Krieger, 2005; Sarfati et al., 2006; Schootman et al., 2010). While these causes are often non-specific in their effects of how they directly influence cancer incidence, they do allow us to conceptualize and measure key factors related inequalities in health. Furthermore, understanding disparities in cancer risk and being able to visualize the place-based differences both in the determinants of cancer inequality can be a valuable tool to both scientist and policy maker alike. The goal of this paper is to investigate the spatial variation in cancer incidence disparities between Hispanic and non-Hispanic populations of the state of Texas between 2000 and 2008 and attempt to identify geographic clusters of disparities in cancer risk between these populations using current incidence data from a population based cancer registry.

Respiratory and digestive system cancers have been identified as often having direct and identifiable causal pathways associated with them, many of which are behaviorally or environmentally influenced. Lung cancer is perhaps the most widely recognized environmentally influenced cancer type, with strong evidence to support the effects of smoking, poor diet and direct inhalation of certain carcinogens including asbestos and other indoor air pollutants (Alberg, Ford & Samet, 2007; Alberg & Samet, 2003; Ruano-Ravina, Figueiras & Barros-Dios, 2003). The exposure to these carcinogens generally leads to errors in somatic cell growth, such as chromosomal abnormalities, cellular mutations, and alterations in tumor suppressor cells. Gastrointestinal system cancers also have a variety of causes, with some consistency between the types of cancer, but other types also have distinct know etiologies. For example, hepatocellular carcinoma (primary liver cancer) has been directly linked with hepatitis infection, alcoholic cirrhosis and dietary aflotoxins (El-Serag, 2012; Stuver & Trichopoulous, 2008) while other digestive system cancers, such as colorectal cancers are heavily influenced by dietary and lifestyle factors (Chao et al., 2005). While the specific etiologies of the cancers of these two body systems sometimes have direct causal paths, they are generally thought to be influenced by both behavioral and environmental circumstances, which interact with familial and genetic pathways in complicated ways.

The state of Texas is the second most populous state in the United States, with a current population estimate of 25.7 million persons. Between 2000 and 2010, Texas was the sixth fastest growing state, and the highest in total numerical population gain (Makun & Wilson, 2011). Additionally, it is consistently in the top five fastest growing states in the nation. The Hispanic population of Texas was estimated to be 10.1 million persons, or over 38% of the population in 2013 and Texas has the second largest Hispanic population, behind only California (Makun & Wilson, 2011). In addition to being a large part of the state’s population, the Hispanic population also faces socioeconomic disadvantages compared to other ethnic groups. The poverty rate for Texas Hispanics was 25.8% according to the 2010 American Community Survey, while non-Hispanic whites only had an 8.8% poverty rate (United States Department of Commerce, 2012). Likewise, Hispanics are more likely to be employed in construction related activities (18.7% compared to 6.1% for non-Hispanic Whites), which could expose this population to more risk from airborne carcinogens.

For such a large and dynamic state, little population-based cancer disparity research has been published for Texas. In a recent study of cancer disparities in Texas counties, Philips et al. (2011) found that an index of socioeconomic well-being was significantly associated with county-level ratios of metastatic to non-metastatic tumors in all-cause, female genital and lung cancers. In a study of El Paso county, Collins et al. (2011) found higher cancer risk for the Hispanic population of that area, and they go on to discuss how in El Paso, areas of the city that had the highest levels of Hispanic population with low levels of education had six times the risk of the more educated areas, and areas with the highest proportion of Hispanic renters had seven times the risk of cancer than other, more socioeconomically advantaged areas. Using a geographically weighted regression approach on data from the Texas Cancer Registry, Tian, Wilson & Zhan (2011) found not only that Hispanics and non-Hispanic Blacks faced disparities in breast cancer mortality, but that these disparities varied over space within the state. These studies likewise point to the placed-based inequality and increased risks that minority groups, including the Hispanic population, face in certain areas within the state. This study will add to the literature on cancer disparities by employing a spatially oriented statistical analysis for the entire state over a more inclusive time period.

With respect to access-based disparities related to cancer risk, Hispanics have been shown to have lower chances of seeking preventative care (Cristancho et al., 2008; Hosain et al., 2011; Lantz et al., 2006; Shih, Zhao & Elting, 2006; Suther & Kiros, 2009) in general, and specifically cancer screening. Reasons for not seeking care include lack of insurance, language barriers and the high cost of health care (Cristancho et al., 2008). In a study of colorectal cancer, Wan et al. (2012) found significant disparities for Hispanics and non-Hispanic Blacks in access to care.

Visualizing disparities across space

From a methodological standpoint, testing for disparities in rates is a relatively straightforward task and a variety of statistical procedures are well suited for it. Specifically, a disparity in two rates can be measured as either a difference in total rates, or as a ratio of risks the groups being compared (Keppel et al., 2005). In terms of visualizing the disparities, this can be more of a challenge. For measuring the disparity between population subgroups, the standardized risk ratio is a useful measure, but it is often subject to noise in the underlying rates, most notably in small populations or in cases of rare disease. Maps of such relative risks, as a result of the noise caused by small populations, often lead to the reporting of unstable risk estimates. Tango (2010) describes a variety of methods for both visualizing and detecting disease clusters. Methods for mapping such risk ratios in a scan-statistic context have been described by Chen and co-authors (2008), and Bayesian disease mapping methods are also cited as being particularly good at mapping spatial disease risk (Anderson, Lee & Dean, 2014; Choo & Walker, 2008; Earnest et al., 2010; Kim & Oleson, 2008; Lawson, 2013; Lawson et al., 2000; Lee & Mitchell, 2014; Lee & Shaddick, 2010). The Bayesian approach allows for smoothing of the relative risk by combining information across spatial units, as well as across time.

It is the purpose of this paper to investigate the spatial variation in cancer incidence disparities between Hispanic and non-Hispanic populations of the state of Texas between 2000 and 2008 using data from a population-based cancer registry. This research adds to the literature in spatial epidemiology by examining the disparities in these two populations over time and space by using a Bayesian modeling methodology, which models the variation in cancer disparities between these two populations within the state. The Bayesian modeling framework is used to specify a series of varying coefficient models as a method of both more accurately modeling the disparity between these two populations, but also for visualizing where the disparities between the populations exist. The goal of this process it to provide a locally accurate depiction of health disparities which state and local health officials could use in combating health inequalities.

Data and Methods

Data source

Data for this analysis come from the Texas Cancer Registry’s (www.dshs.state.tx.us/tcr/) Limited-Use data file from 2000 to 2008. Access to these data was approved by the Texas Department of State Health Services IRB #12-030. These data consist of de-identified individual records of primary cancer diagnoses by oncologists in the state of Texas. For the purposes of this study, relevant variables in the data include year of diagnosis, age, sex, Hispanic ethnicity, International Catalog of Disease for Oncology (ICD-O-3) codes for cancer diagnosis site and county of residence at the time of diagnosis. Two main types of cancer were chosen: digestive system (ICD-O-3 codes C150–C488) and respiratory system cancers (codes C300–C399). These cancers were chosen because several of the sub-types of these cancers have been linked to environmental or behavioral influences, and several have also been shown to vary between ethnic groups in their incidence (Howe et al., 2006; Singh & Hiatt, 2006; Singh & Siahpush, 2002; Wiggins et al., 1993; Willsie & Foreman, 2006). These two cancers are selected for study, because they constitute 41% of all cancers in the state for this period. For the years of this study a total of n = 155,652 digestive and n = 124,438 respiratory system cases were in the data. The most prevalent form of digestive system cancer was colorectal cancer, with 53% of digestive cancers, and squamous cell carcinoma of the lung was the most prevalent respiratory cancer, representing 22% of all cases. The distributions of cancers by specific location are provided in Table 1.

Table 1 Distribution of cancers by system and type.

Cancer type	Count	Percent	
Digestive cancers			
Gum and mouth	506	0.3	
Esophagus	7,745	5.0	
Stomach	14,190	9.1	
Small intestine	4,183	2.7	
Colon and rectum	85,821	55.1	
Anus and anal canal and anorectum	2,876	1.8	
Liver	14,032	9.0	
Gallbladder	2,095	1.3	
Other biliary	2,702	1.7	
Pancreas	19,124	12.3	
Retroperitoneum	780	0.5	
Peritoneum, omentum and mesentery	1,026	0.7	
Other digestive organs	572	0.4	
Respiratory cancers			
Nose, nasal cavity and middle ear	1,469	1.2	
Larynx	7,720	6.2	
Lung and bronchus	113,357	91.1	
Pleura	1,295	1.0	
Trachea, mediastinum and other respiratory organs	596	0.5	

There are thirteen different types of site-specific cancers under the digestive system and five site-specific cancers under the respiratory system, according to ICD-O-3 designations within the data. Among the digestive system cancers, colon and rectum cancer was the most prevalent, at 55.1% of all cases, and lung and bronchus cancer was the most prevalent for respiratory system cancers, with 91.1% of all cases.

There are two dependent variables in this analysis, and they represent the count of either digestive or respiratory cancers in each of the 254 counties of Texas between 2000 and 2008. The data are stratified by ethnicity into two categories Hispanic and non-Hispanic. The stratification of the cases is accomplished by using the Hispanic ethnicity variable in the registry. This variable was very complete in the data, and was only missing for 1.4% of cases. Thus for each year, there are two separate counts for each cancer type and for each of the 254 counties in the state. Since the dependent variables are counts, they are generally expressed as a standardized ratio of counts to expected counts. This is typically called the standardized incidence ratio (SIR), and is expressed: SIRijk=yijk/eijk

where yijk is the count of cases in the ith county for the jth year for the kth ethnicity and eijk is the expected number of cases in the county for each group. Here, to estimate the expected number of cases for each county, year and ethnicity, an assumption of equal risks is used. The expected number of cases in each county, year and ethnicity, eijk, is calculated by assuming each county has the average incidence rate for each ethnicity (Hispanic and non-Hispanic) for the whole state for the period 2000 to 2008, or: eijk=Σnijk∗rk,

where nijk is the number of residents in each county for each ethnicity, and rk is the average incidence rate for the state, for ethnicity k, for the period 2000 to 2008. This is repeated for each type of cancer: digestive and respiratory. This generates a set of expected values for the Hispanic and non-Hispanic population of each county, using the statewide rate for each ethnic group and the county population size for each group.

To control for background characteristics of the counties, and to measure proxies for factors affecting cancer risk, four independent variables are constructed. The first of these is the metropolitan status of the county, which is measured as a dummy variable indicating whether the United States Department of Agriculture’s Economic Research Service considers the county metropolitan. Metropolitan counties are coded as 1, and non-metro counties are coded as 0. The poverty rate in each county is calculated from the US Census Bureau’s Summary File 3 for 2000, and is expressed as the proportion of all residents living below the poverty line in 1999. The proportion of the labor force in construction is used to measure a crude proxy for occupational exposure to certain carcinogens. This is again measured using the Census’s Summary File 3 and expressed as a proportion. Finally, the Area Resource File (US Department of Health and Human Services, 2009) for 2008 is used to measure the number of hospitals in each county per 10,000 residents. This is used as a crude proxy for healthcare access in each county.

Statistical methods

Model specification

Since the dependent variable is a count, a Poisson distribution is used to model the outcome. To model this outcome, a log-linear Poisson hierarchical regression model for each county, i, year, j, ethnicity, k, and type of cancer, C, is specified as: yCijk|θCijk∼PoissoneCijk*θCijk.

The relative risk function, θCijk, can be parameterized using a number of different models, the present paper considers a Bayesian model specification.

In the Bayesian modeling paradigm, all model parameters are considered to be random variables and are given a prior distribution. All inference about these parameters is made from the posterior distribution of these parameters, given the observed data and the information given in the priors. This is generally referred to as Bayes Theorem, and typically stated as: pθ|y∝py|θpθ.

Where p(θ|y) is the posterior distribution of the model parameter of interest, p(y|θ) is the model likelihood function, here defined as a Poisson likelihood, and p(θ) is the prior distribution for the parameters in the model. Inference for all parameters is done via their posterior distribution, which can be used to derive mean values, quantiles or other descriptive statistics. One useful method for summarizing these distributions is the Bayesian Credible Interval (BCI), which is not unlike a traditional frequentist confidence interval, which gives the values of the posterior density for each parameter that contain 100∗(1 − α)% of the posterior density. Inference on these BCI regions usually consists of examining if the null hypothesis value of the parameter, typically zero, is contained in the interval.

Since the primary interest in this paper is the relative difference between the incidence of cancer in the Hispanic and non-Hispanic populations of each county, the simplest way to parameterize the model is as a linear difference in the incidence rates, conditional on the background spatio-temporal random effects. This is the first model considered, and is parameterized as: (Model 1) lnθCijk=αC+δC*ethCi+∑kβCkxik+uCi+vCi+tCj+ψCijαC∼U−inf,infδC∼N0,.0001βCk∼N0,.0001vCi∼N0,τCvuCi∼N1nj∑j∼iuCj,τCu/nitCj∼N0,τCtψCij∼N0,τψC,

which follows the standard form for spatio-temporal disease incidence models commonly used in the literature (Blangiardo & Cameletti, 2015; Blangiardo et al., 2013; Held et al., 2006; Knorr-Held, 2000; Lawson, 2013; Lee & Mitchell, 2014; Schrodle & Held, 2011b; Ugarte et al., 2009). This model specifies the relative risk as a linear function of a grand intercept for each cancer type, αC, a mean difference between the two ethnicities (eth) for each cancer type. Here, it is important to note the eth variable is binary, with 1 indicating the Hispanic rate, and 0 representing the non-Hispanic rate, or the reference group. δC, is a linear predictor effect of the independent variables for each cancer type, ΣβkC xik, a “convolution” spatial prior, corresponding to the Besag, York & Mollie (1991) model, which incorporates an unstructured heterogeneity term for each county and cancer type, vCi, and a correlated heterogeneity term specified as a conditionally autoregressive random effect, uCi, a temporally unstructured random effect for each year and cancer type, tCj1 and finally a spatio-temporal interaction random effect, ΨCij, which follows the Type 1 specification in Knorr-Held (2000). In this model there is a single parameter (δ) for measuring the disparity between Hispanics and non-Hispanics for each cancer type, and this is done on average for the entire state. This model additionally captures the underlying characteristics of the counties, the overall spatial structure of cancer risk, and the temporal variation between years in the relative risk. Priors are assigned to all parameters in a minimally informative fashion, with an improper flat prior for αC, high variance Normal distribution priors for the δC and βC and vCi, a Normal distribution prior for tj and vague Gamma priors for the precisions of the unstructured heterogeneity, correlated heterogeneity, temporal and spatio-temporal components. For all models, the Normal distribution priors are specified in terms of their mean and precision, which is common in Bayesian modeling, with the precision being the inverse of the variance: τ = 1/σ2, such that low precisions equal high variances.

A second model adds more flexibility to Model 1 by including a random slope for each county’s difference between Hispanic and non-Hispanic risk. This model is specified as: (Model 2) lnθCijk=αC+δCi*ethCi+∑kβCkxik+uCi+vCi+tCj+ψCijαC∼U−inf,infδCi∼δC0+δCi,δCi∼N0,τCδβCk∼N0,.0001vCi∼N0,τCvuCi∼N1nj∑j∼iuCj,τCu/nitCj∼N0,τCtψCij∼N0,τψC

which is similar to (1), but includes a δCi term which allows the differences between Hispanic and non-Hispanic risk to vary between counties, instead of assuming the difference between the two population is the same across the state, and is equivalent to an unstructured random-slopes model for the disparity. This is much like the spatially varying coefficient model discussed elsewhere (Banerjee, Carlin & Gelfand, 2004; Gelfand et al., 2003), except in this model, the random slope term is not spatially correlated.

A final model adds a correlated slope for the disparity parameter to Model 2. This model follows the example of previous authors, who model the disparity between groups as a spatial conditionally autoregressive random slope (Tassone, Waller & Casper, 2009; Wheeler, Waller & Elliott, 2008). This model has the form: (Model 3) lnθCijk=αC+δCi*ethCi+∑kβCkxik+uCi+vCi+tCj+ψCijαC∼U−inf,infδCi=δC0+δCi,δCi∼N1nj∑j∼iδCj,τCδ/niβCk∼N0,.0001vCi∼N0,τCvuCi∼N1nj∑j∼iuCj,τCu/nitCj∼N0,τCtψCij∼N0,τψC,

which smooths the disparity parameter over neighboring counties within the state.

Clustering in risk

One of the goals of this analysis is to identify areas where the disparity in risk between these population subgroups is clustered. To identify clusters of risk for Hispanics, relative to non-Hispanics, Bayesian exceedence probabilities are used (Lawson, 2013). An exceedence probability is: PrθCijk>θ*,

where θ* is some critical level of risk that is specified. Here, the exceedence probability of the Hispanic rate being 25% higher (θ* > 1.25) than the non-Hispanic rate is used. These exceedence probabilities will allow the “significance” of the disparity to be mapped. When the probability is high, then there is a statistically important difference between the risk in the Hispanic and non-Hispanic cancer incidence, and that the area represents a spatial cluster of risk.

For geographic modeling, neighbors are identified using a first order Queen contiguity rule. Other neighbor specifications were examined, specifically a first order rook contiguity rule, and the results were substantively robust to this other neighbor specification. Also, since the precision terms for Bayesian hierarchical models have been shown to be sensitive to prior specifications, a sensitivity analysis is performed. The models specified above all considered proper Gamma (.5, .0005) priors for all precision terms, and to gauge the sensitivity of the results, Uniform distributions for the precisions are also considered. These prior distributions have been used by other authors, and are thought of to be a sufficiently vague prior for the precision for these parameters.

Computing—INLA

The software R (R Development Core Team, 2015) and the R package R-INLA (Martins et al., 2013; Rue, Martino & Chopin, 2009) were used to prepare data for analysis and parameter estimation. The Integrated Nested Laplace Approximation, or INLA, approach is a recently developed, computationally simpler method for fitting Bayesian models (Rue, Martino & Chopin, 2009), compared to traditional Markov Chain Monte Carlo (MCMC) approaches. INLA fits models that are classified as latent Gaussian models, which are applicable in many settings (Martino & Rue, 2010). In general, INLA fits a general form of additive models such as: η=α+∑j=1nffjuij+∑k=1nββkzki+εi,

where η is the linear predictor for a generalized linear model formula, and is composed of a linear function of some variables u, β are the effects of covariates, z, and ε is an unstructured residual (Rue, Martino & Chopin, 2009). As this model is often parameterized as a Bayesian one, we are interested in the posterior marginal distributions of all the model parameters. Rue & Martino (2007) show that the posterior marginal for the random effects (x) in such models can be approximated as: p˜xi|y=∑kp˜xi|θk,yp˜θk|yΔk

via numerical integration (Rue & Martino, 2007; Schrodle & Held, 2011a; Schrodle & Held, 2011b). The posterior distribution of the hyperparameters (θ) of the model can also be approximated as: p˜θi|y∝px,θ,yp˜Gx|θ,y|x=x*θ,

where G is a Gaussian approximation of the posterior and x*(θ) is the mode of the conditional distribution of p(x|θ, y). Thus, instead of using MCMC to find an iterative, sampling-based estimate of the posterior, it is arrived at numerically. This method of fitting the spatio-temporal models specified above has been presented by numerous authors (Blangiardo & Cameletti, 2015; Blangiardo et al., 2013; Lindgren & Rue, 2015; Martins et al., 2013; Schrodle & Held, 2011a; Schrodle & Held, 2011b), with comparable results to MCMC.

To summarize the posterior distributions of the model parameters, posterior means and 95% credible intervals are calculated. Three models specified in ‘Model specification’ were examined. Model fit and improvement is assessed between the models with the Deviance Information Criterion (DIC) (Spiegelhalter et al., 2002). The DIC measures the penalized deviance of each model, with the penalty term representing the model’s estimated number of parameters. DIC for the INLA models is described in Rue, Martino & Chopin (2009) and uses the model deviance Dθ=−2logpy|θ+pD,

plus a penalty component, pD, which is an approximate number of parameters in the model. DIC is used, here, as a measure of relative model performance, and models with lower DIC values are preferred over those with higher DIC, analogous to the standard AIC criteria.

Results

Descriptive results

Descriptive statistics for the dependent variable and the predictors are presented in Table 2.

Table 2 Descriptive statistics for dependent and independent variables used in the analysis.

Cancer type and year	Mean # cases	IQR	Mean # cases (non-Hispanic)	Mean # cases (Hispanic)	Mean SIRH/SIRNH	
Digestive cancer cases per county	
2000	30.9	18	49.9	12.0	0.87	
2001	32.2	18	51.8	12.6	1.44	
2002	32.9	19	52.6	13.2	1.18	
2003	33.7	19.25	53.5	14.0	1.14	
2004	34.4	22	54.0	14.8	1.31	
2005	34.8	22	53.9	15.8	1.32	
2006	35.2	21	54.3	16.1	1.30	
2007	36.1	23	55.8	16.4	1.46	
2008	36.1	20	55.1	17.0	2.06	
	155,652 total cases					
Respiratory cancer cases per county	
2000	25.6	15	46.0	5.2	1.28	
2001	26.5	17	47.2	5.8	1.42	
2002	26.9	17	48.2	5.6	1.16	
2003	27.8	17	49.4	6.1	1.62	
2004	27.6	16.25	49.2	5.9	1.18	
2005	28.1	17	49.9	6.4	1.48	
2006	27.4	16	48.4	6.5	1.67	
2007	27.8	16	48.7	6.8	1.61	
2008	27.2	15	48.1	6.4	1.54	
	123,437 total cases					
Predictors	Mean	IQR				
% in poverty	17.76	6.58				
Hospitals/10,000 people	0.66	0.79				
% in construction	8.11	3.15				
% metro counties	30.31	1.00				
Notes.

n = 254 counties.

Table 3 Results for the alternative Bayesian model specification parameters.

	Model 1	Model 2	Model 3	
Parameter	Posterior mean (95% credible interval)	Posterior mean (95% credible interval)	Posterior mean (95% credible interval)	
	Digestive	Respiratory	Digestive	Respiratory	Digestive	Respiratory	
α	−.081 (−.119– −.043)	−.066 (−.095–−.037)	−.098 (−.137–−.059)	−.074 (−.103– −.044)	−.097 (−.136–−.057)	−.074 (−.103– −.044)	
β							
% in Poverty	−.031 (−.052– −.010)	.002 (−.027–.033)	−.034 (−.057–−.011)	.001 (−.031–.032)	−.033 (−.057 –.010)	.001 (−.032–.030)	
Hospitals per capita	−.016 (−.037–.004)	−.007 (−.032–.016)	−.015 (−.037–.005)	−.008 (−.033–.016)	−.016 (−.037–.005)	−.007 (−.032–.018)	
% in Construction	−.011 (−.027–.005)	.050 (.028–.072)	−.009 (−.026–.008)	.050 (.027–.072)	−.001 (−.026–.008)	.050 (.028–.073)	
Metro County	.023 (−.009–.056)	.052 (.007–.095)	.023 (−.011–.057)	.054 (.009–.099)	.021 (−.011–.056)	.054 (.009–.099)	
Hispanic disparity, δ	.052 (.038–.066)	.107 (.087–.126)	.138 (.106–.171)	.146 (.109–.184)	.152 (.122–.183)	.152 (.112–.192)	
Model fit							
Deviance (D¯)	21256.2	18625.7	20790.2	18462.5	20775.6	18436.8	
DIC	21630.2	19004.4	21240.7	18888.5	21217.2	18859.9	
pD	373.9	378.7	449.9	426.0	441.6	423.1	
Hyperparameters							
τt	477.8	1552.5	478.6	1546.5	478.0	1538.8	
τu	331.3	555.6	432.3	898.1	428.7	923.1	
τv	133.9	24.2	93.7	20.4	92.6	20.8	
τ δ	–	–	52.3	67.5	15.6	17.9	
τ ψ	297.1	284.8	296.2	287.3	296.5	288.7	
Notes.

Parameters in bold type represent estimates whose credible intervals do not contain 0.

A gradual increase in the average number of cases per county is observed over the nine years of data. Also, many more cases of both types of cancer (on average) occur to non-Hispanics than to Hispanics. It should be noted that between 25% (2005) and 36% (2000) of counties had a zero count for Hispanic digestive cancer cases and between 38% (2003) and 46% (2002) had a zero count for Hispanic respiratory cancer cases. 2 Also presented in Table 2 are the observed average risk ratios for the state for each year. These are calculated as ratio of the observed SIR for Hispanics (SIRH) and the observed SIR for non-Hispanics (SIRNH) for each year. For digestive cancers, every year shows an elevated risk for Hispanics compared to non-Hispanics, and all years except 2000 show an elevated risk of respiratory cancer for Hispanics. Likewise, respiratory cancers show a consistent trend of higher risk in Hispanics, but not as high as for digestive cancers. With respect to the predictor variables, in 2000 nearly 18 percent of the population of Texas was in poverty, with a wide degree of variation as seen by the inter quartile range. On average there were .66 hospitals per 10,000 people in each county in the state, and there were sixty-five counties with no hospitals. Slightly over 8 percent of the work force was employed in construction, and the USDA considered thirty percent of counties in the state to be metropolitan.

Results of Bayesian models

Table 3 presents the posterior means of the regression effects for the fixed effects in the three models described above. Also, 95% Bayesian credible intervals are provided for each parameter. Model DIC values are also provided at the bottom of the table for each model. Lastly, summaries for the model hyperparameters provided.

Across the three models, some of the fixed predictors show similar patterns. For digestive cancers, the poverty rate shows a negative association with overall cancer risk in Models 1 through 3. This suggests that in areas of higher poverty, the average cancer risk is lower. Respiratory cancer incidence is affected consistently by two of the predictors. The proportion of the work force in construction is positively associated with respiratory cancer risk in the three of the models, potentially suggesting an occupation-specific risk pattern. Likewise, a metropolitan disadvantage is seen, with higher total cancer risk in metropolitan areas. Both of these variables are in line with expectations in terms of respiratory cancer risk.

When the three models are compared using the DIC, Model 3 shows the best model fit for each cancer type, with the DIC being lowest for this model. Strong evidence is present that Model 1 is not adequate to describe the patterns of Hispanic/non-Hispanic disparities in either cancer, as every other model shows large drops in DIC. When comparing Models 2 and 3, strong evidence also exists for adding the spatially correlated random slope term, again with a large drop in DIC.

Turning to the Hispanic disparity parameters, in all models, there persists an overall average disparity between Hispanics and non-Hispanics, with the former consistently showing elevated risk for both types of disease, net of the ecological factors, and the random effects. For digestive cancers, we see an increase in risk (eδ) between 5.3 and 16.4 percent, on average and between 3.8 and 20 percent when considering the 95% credible intervals, depending on the model. For respiratory cancers, we see an increase between 11.2 and 16.4 percent on average, and 9.1 and 21.1 percent when examining the credible intervals. For Models 2 and 3, the coefficients of the models are best presented graphically, as each county has an estimate for the disparity for each cancer type. These estimates are presented in Fig. 1 as posterior mean estimates of the Hispanic disparity in relative risk (eδC) for each county for Models 2 and 3.

Figure 1 Hispanic relative risk from Models 2 and 3.

The first column of Fig. 1 shows the Hispanic disparity random effect from Model 2, for respiratory and digestive cancers, respectively, when the disparity parameter was treated as unstructured. The second column of the figure shows the same parameter, when it was treated as a spatially structured random effect (Model 3). For both respiratory and digestive system cancers, Hispanics show elevated risk in the eastern portion of the state, but they also show elevated risk in the central portion of the state for digestive system cancers, but not for respiratory cancers. The value of these figures is that the actual disparity in risk is being visualized, which shows us where within the state public health officials might try to focus activities in order to reduce the disparity in risk between these two populations.

Spatio-temporal relative risk estimation

Figure 2 displays the estimated Hispanic relative risk for digestive cancers (eθ) for each year, 2000 to 2008, estimated from Model 3.

Figure 2 Hispanic fitted SIR from 2000 to 2008—digestive cancers.

The quantity being mapped is the linear predictor of the Poisson distribution (eθ), with all random effects included, which is interpreted as the model-based standardized incidence ratio (SIR). Each panel in the figure shows the spatial distribution for each year between 2000 and 2008. We see a general concentration of elevated Hispanic digestive cancer risk in the eastern portion of the state, as evidenced by relative risks greater than one (darker blue in color). This pattern is consistent, if not increasing over time, with more counties showing greater Hispanic relative risk over time. Lower risk (eθ < 1) for Hispanics occurs in North and Western Texas, and also along the border with Mexico, except for a few counties in extreme South Texas in the latter time periods.

Figure 3 provides the complementary space–time risk map for the respiratory cancer outcome. Again, we see higher Hispanic risk in Eastern Texas, but perhaps a more concentrated pattern, compared to the digestive cancer maps. Also present is the lower risk in North and West Texas, as seen in Fig. 2 for digestive cancers. Figure 3 also highlights a consistent spatial cluster of high risk in extreme East Texas for a cluster of three to five counties located North of Harris county (city of Houston). These counties include Montgomery, Liberty, San Jacinto, Walker, Polk and Orange. These counties are quite rural and have low proportions of Hispanic residents (average of 9.3%, or about 8,900 Hispanic persons on average per county).

Figure 3 Hispanic fitted SIR from 2000 to 2008—respiratory cancers.

For each cancer type, Figs. 4 and 5 illustrate the exceedence probabilities for the Hispanic disparity parameter from Model 3.

Figure 4 Exceedence probabilities for digestive cancer clusters.

Figure 5 Exceedence probabilities for digestive respiratory clusters.

Figure 4 shows the exceedence probabilities for the digestive cancers over time. There is a persistent, significant, meaning a Pr(θ > 1.25) between 95 and 100% in the eastern portion of the state on the Louisiana border, with smaller areas of isolated significant risk throughout the state, which appear to emerge over time, versus being consistent across time.

Similarly, the disparity in respiratory cancer risk clusters in the same areas as for digestive risk, with an apparent secondary cluster in the northeastern areas of the state. Both of these clusters are persistent across time.

Finally, a sensitivity analysis of alternative priors for the model hyperparameters (all τ’s) showed very close agreement between the vague Gamma (.5, .0005) and the flat prior distributions. Since Model 3 showed evidence of being the best fitting model, the sensitivity analysis focused on its estimates. The precision point estimates for the temporal random effects (τt) for the digestive and respiratory cancers, respectively were 478.0 and 1538.8 from the Gamma prior and 441.5 and 1822.5 from the flat prior. The precisions for the uncorrelated heterogeneity (τu) were 428.7 and 923.1 for the Gamma prior and 354.0 and 1095.8 for the flat prior. The precisions for the correlated heterogeneity (τv) were 92.6 and 20.8 for the Gamma prior and 92.5 and 19.9 for the flat prior. The precisions for the varying disparity parameter were 15.6 and 17.9 from the Gamma and 14.9 and 17.0 from the flat prior. The precisions for the spatio-temporal random effect (τψ) were 296.5 and 288.7 for the Gamma prior model and 298.3 and 283.8 for the flat prior model. While this is only one model, the overlap between the precisions is strong enough to validate the results. The one notable difference is the random effect for the unstructured heterogeneity (τu), which showed a lower precision (higher variance) in the Gamma prior model, although the parameter’s 95% credible interval did show significant overlap between the two prior specifications (Fig. 6).

Figure 6 Marginal densities for model hyperparameters.

Discussion

This paper illustrated the application of the Bayesian varying coefficient models to the study of cancer incidence disparities between the Hispanic and non-Hispanic population of Texas over the period 2000 to 2008. This paper adds to the literature in health disparities within the state of Texas by using advanced Bayesian statistical methods to investigate the spatial non-stationarity of health disparities in two major form of cancer incidence. The primary goal of this analysis was to investigate the spatial variation in cancer incidence disparities between Hispanic and non-Hispanic populations of the state of Texas between 2000 and 2008 and attempt to identify geographic clusters of disparities in cancer risk between these populations using a spatially varying coefficient model (Banerjee, Carlin & Gelfand, 2004; Gelfand et al., 2003; Tassone, Waller & Casper, 2009; Wheeler, Waller & Elliott, 2008). A Bayesian modeling framework was used, using a variety of model specifications, including models that included interactions between space and time. Alternative model specifications modeled the disparity in incidence between the two subpopulations differently, from a fixed effect on the grand mean to a spatially varying coefficient model for each county in the state. The flexibility of the Bayesian framework also allowed for the models to be compared using standard model complexity criteria (DIC).

The model that best fit the data was the space–time model with a spatially varying slope for the disparity between Hispanics and non-Hispanics, according to the minimum DIC criteria. This suggests that the disparity between Hispanics and non-Hispanics in these two cancer types is best modeled through a spatially structured model, which allows for spatially structured variation in risk. This also suggests that there are counties within the state where the Hispanic population is at higher risk for both of these cancers, and that these counties typically occur closely to one another spatially.

Overall, a general disparity in terms of both cancers for Hispanics was found, where they face higher risk for both digestive and respiratory cancers than the non-Hispanic population of the state. Significant effects were found on cancer-specific risks consistently including the county poverty level, metropolitan status of the county and the proportion of the workforce in construction. The labor force composition finding makes sense, as workers in construction industries often face higher levels of exposure to airborne particulates that could increase cancer risk. The finding for the county poverty rate was that in areas with higher poverty, the overall relative risk of cancer was lower, and deserves more discussion. This effect was seen for both cancer types, in all but the final model (Model 3), and is in stark contrast to findings from national data (Singh et al., 2003) for many types of cancer, which show higher incidence and mortality in both Hispanics and non-Hispanics in areas with higher poverty. Singh et al. (2003) did not use data from Texas, and the time period for the present study is later than those considered in their report. It is possible that the experience of the Texas population is different from the data used in their study; such local variations are common in health research.

Significant spatio-temporal clusters of excess risk for the Hispanic population were found in the eastern portion of the state for both cancer types. These clusters focused around a small group of rural counties in Eastern and Northeastern Texas. These counties are generally located north and east of Harris county (city of Houston). For digestive cancers, clustering begins in Jasper, Liberty, Orange and Walker counties, and spreads over time to include other neighboring counties. For respiratory cancers, a similar area is covered, but also includes Bowie, Gregg, Henderson and Smith counties in northeastern Texas. These counties are quite rural and have low proportions of Hispanic residents (average of 8.5% in 2000, or about 7,450 Hispanic persons on average per county).

This study had several limitations. First, the cancer incidence data had no information on residential histories of the individual cases. Any environmental exposure that could have influenced cancer risk may have come from a previous residential location. Unfortunately, the cancer registry data used in this study had no information on this subject. Secondly, this was an ecological study, and no individual level covariates (besides Hispanic ethnicity) were used, and the proxy measures of environmental exposure (metro status and proportion in construction) are crude measures, and better measures could be included in future work. Thirdly, this study lumped a wide array of specific cancer sites together (see Table 1) into two broad body “systems” for the analysis. This was done to avoid cases of extremely small counts, and more information could be gained by considering more site-specific cancers.

Further research is needed to investigate the specifics of the counties identified in the analysis as having excess Hispanic cancer risk. This can be done by a more localized analysis of the individual-level data this analysis is derived, and by investigating housing conditions, access to healthcare and potential environmental contaminants in these areas directly. Such ecological analyses as that presented here are rarely truly informative for individual cancer diagnoses, but they can be very influential in terms of public health activities to reduce cancer disparities at the population level.

Supplemental Information

Supplemental Information 1 Simulated data and the code to fit the models

Click here for additional data file.

Additional Information and Declarations

Competing Interests

Author Contributions

Ethics

Data Availability

1 Other prior distributions, including a first order random walk (RW1) priors were used, but did not increase model fit in this case, so the simpler exchangeable random effect for time was used in the final model.

2 The large number of zeros in the data suggests that a zero-inflated distribution be used as the model likelihood. A zero-inflated Poisson model was considered for the analysis (results available from the author), but the DIC of said models suggested the Poisson model fit the data better.

The author declares there are no competing interests.

Corey Sparks conceived and designed the experiments, performed the experiments, analyzed the data, contributed reagents/materials/analysis tools, wrote the paper, prepared figures and/or tables, reviewed drafts of the paper.

The following information was supplied relating to ethical approvals (i.e., approving body and any reference numbers):

Texas Department of State Health Services IRB #12-030.

The following information was supplied regarding data availability:

The original data for this study is considered restricted by the Texas Cancer Registry and is made accessible via a signed data use agreement, only after the study protocol has been approved by the Texas Cancer Registry IRB review process. Therefore I have provided simulated data from the models considered and the code to fit the models in Supplemental Information 1.

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
