# Peer review of "An examination of disparities in cancer incidence in Texas using Bayesian random coefficient models"

_PeerJ, doi:10.7717/peerj.1283_

## Round 0.1 · original submission · Major Revisions

· Academic Editor

Major Revisions

Please respond to the reviewers comments in particular the use of English and the clarifications that they are seeking

Yours,
Prof D Gareth Evans

Reviewer 1 ·

Basic reporting

please see "Gneral Comments"

Experimental design

please see "Gneral Comments"

Validity of the findings

please see "Gneral Comments"

Additional comments

The statistics in this paper are basically sound and reasonably well expplained but the statistical methoology is not or itself of novel interest. I would be happy for this paper to be published providing that the editor feels that the results obtained are of interest in themselves (as somone without knowledge of health inequalities I can't comment on this), or that the use of these methods is novel within the field There are some points I don't understand and I'd be grateful if the author could clear them up:

1. page 4 "although since a linear regression was used, no relative risks are available" What does this mean?

2 I dont understant why r in the equation on page 7 needs an "ijk" suffix since in is an average over all countes, racial groups and
years.

3. "high variance Normal distribution priors": please explain in the text somewhere around Model 1 that normal distributions are being written using the precision notation and explain what this means. Explicit priors for the precisions should be added to all the models.

4. In all the models should beta_{C} have a a multivariate prior as it is a vector? It appears to be a vector rather than a scalar.

4. a_{C} ~ flat not a_{C} = flat in models 2-4 (acutally I beleive
U(-infinity,infinity) is a more standard notation for a non-informative prior.

6. In models 3 and 4, should a psi_{Cij} term be aadded to the first line?

7It seemse a shame that the hispanic disparity (HD), the main feature of interest in this study, cold not be put in the table of results for models 2 - 4, nor an "oveall" estimate of HD given for these modles. Would it be possible to parameterise the spatially
varying HD as (deltabar + delta_{Ci} ) * eth_Ci} and then reprting the posterior for deltabar for models 2-4. (If this is too much extra work don't worry about it, as obviously it would involve rewriting (slightly) the three models 2-4 and re-running them, but I do think it would be nice to have an answer to the basic question "does ethnicity affect cancer counts" for all 4 models)

8 Is G in the DIC definition the namber of MCMC samples?

9. The paper needs careful re-reading for English and gramm ar
For example:
delete "are" in sentence beginning "Maps of such relative risk .." on page 5
delete "to estimate" at beginning of the sentect just above the second equation on page 7 and insert "year and ethnicity" after "county"
insert "a" before "dummy variable" at top of page 8

delete "for digestive cancers" in sentence begining "It should be noted that..." in section 3.1

Reviewer 2 ·

Basic reporting

This manuscript needs a careful copy editing, as there are numerous grammatical and typographical errors. The paper is sloppily written, particularly in the mathematical notation.

Model Specification: Model 2 is not like a typical spatially varying coefficient model in the cited papers because the varying effects here for ethnicity are exchangeable, which is a simpler structure than the published models. This difference should be noted.

Model Specification: “While this model itself is not new, the application of it to a health disparities outcome is new contribution.” There are many published examples of models with spatially varying random effects used to estimate racial/ethnic health disparities. Examples include Wheeler et al. (2008) and Tassone et al. (2009). A shared component model is a common choice for modeling differences in disease risk for two or more groups simultaneously. The present paper is somewhat lacking in relevant statistical references.

References:
Tassone , Waller, Casper. 2009. Small-area disparity in stroke mortality. Epidemology, 20(2).
Wheeler, Waller, Elliott. 2008. Modeling epilepsy disparities among ethnic groups in Philadelphia, PA. Statistics in Medicine, 27(20).

Experimental design

Data Source: Why are digestive system (and respiratory system) cancers lumped together when there is likely heterogeneity in risk factors? Why not create models for each cancer site? Even cancers of the same type have heterogeneity in risk factors for sub-types (e.g., non-Hodgkin lymphoma). This would be more informative and would be an improvement to this manuscript.

Data Source: To what extent are small numbers a problem for the SIR strata? Stratifying by ethnicity, county, and year could lead to very small counts. Are there strata with zero counts or zero expected counts? If so, how are these handled?

Data Source: The description of the expected counts e_ijk does not match the notation, as the description states that the rate for the period 2000 to 2008 was used, but the notation for the rate is r_ijk.

Model Specification: It would be helpful for interpretation to explicitly write the formula linking theta_Cijk to the observed data.

Model Specification: It is not clear how in Model 1 how delta_C is “a mean between the two ethnicities for each cancer type”. You should specifically define the variable eth_Cik. It is hard to interpret this model term without a clear definition of the variable.

Model Specification: State that the prior distributions are defined using precision.

Model Specification: Tau_deltaC does not have a prior distribution definition in Model 2.

Validity of the findings

Results of Bayesian models: “Interestingly, Model 4 shows no significant effects of the predictor variables. This suggests that the predictors used may just be proxies for the spatial concentration in risk measured by the spatial random effect in Model 4.” It is difficult to separate trend from spatial autocorrelation. Another possible conclusion is that there is spatial variation in the covariates that is related to the trend in relative risk, but the spatial random effects explains some of this variation.

No limitations of this study are discussed. A major limitation is the lack of residential histories for considering temporal lags in environmental exposures, particularly because the cancer types were selected because they are associated with environmental risk factors.

Reviewer 3 ·

Basic reporting

A well written and clear paper.

Experimental design

There are a variety of spatial analysis techniques that have been developed over recent years ranging from binomial kriging to the Bayesian methods employed here. Each has their own strengths and weaknesses. That said the methods employed in this paper represent a credible approach to the dataset considered.

Validity of the findings

Findings are robust within the Bayesian approach adopted.

Additional comments

The article is well structured and well described both in terms of background and methodology. The area of health inequality is important. The article has clear conclusions from their results could be used to form policies.

---

## Round 0.2 · Major Revisions

· Academic Editor

Major Revisions

I am afraid one of our reviewers still has serious concerns. Please would you address these and correct the grammatical errors in any revision.

Reviewer 2 ·

Basic reporting

1. The goal of the paper is stated in the Introduction is to “identify geographic clusters of disparities in cancer risk”. However, the paper never identifies any statistically significant clusters of cancer risk disparities. In the Discussion, it is stated that “The primary goal of the analysis was to examine the usefulness of the spatially varying coefficient model.” What really is the goal of this paper? Is this goal achieved? If it is the first statement, then I would say that this goal is not adequately addressed.
2. There are still some areas where the statistical model details could be clarified. See specific comments below.
3. The limitations are not adequately discussed. See comments below.
4. The figures (maps) need revising.
5. This paper still needs a careful copyediting, as I found several grammatical errors.

Experimental design

1. Figure 1: More than four colors would be an improvement. Which color corresponds to a relative risk of 1? It appears to be too colors from the legend.
2. Figure 2 and 3: Reverse the color ramp to be consistent with Figure 1 and convention.
3. Data source: How complete is the Hispanic ethnicity variable in the registry data? Is it complete for all cases? If not, what percent are missing?
4. Data source: Why is the grand state rate used to calculate the expected cancer counts? Why is year not considered? Why is ethnicity not considered?
5. Model specification: The relative risk symbol on the left hand side in the models needs subscripts.
6. Model specification: The eth term should be very clearly stated to be a binary variable with a specific reference level.
7. Model specification: Model 2 would be clearer with the delta term having a subscript i.
8. Model specification: The cited papers related to model 3 used versions of a conditionally autoregressive prior. The term spatially autoregressive has a specifically different meaning.

Validity of the findings

9. Results: Is the “cluster” in Figure 3 statistically significant?
10. Limitations: There are many limitations (not just one) and they should be stated. There is heterogeneity in risk factors for the cancers that are grouped to form the outcome variables (digestive and respiratory cancers). The cancer sites that form each group should be listed in a table along with the count and percent that each site accounts for of the total cancer cases to be more transparent. The surrogate for occupational exposure to carcinogens is crude. What specific carcinogens is this surrogate thought to represent? Is it more relevant for digestive or respiratory cancers? No surrogate is used for environmental exposures. The study is ecological and the covariates are not for individual-level data. Hence, no individual measures of exposure are considered.

---

## Round 0.3 · Minor Revisions

· Academic Editor

Minor Revisions

You have very adequately responded to the reviewers concerns

Your edits have unfortunately deleted spaces between words please correct in abstract and text

Please insert 1.4% missing value for Hispanic

Please correct spelling of Hispanic(e)

---

## Round 0.4 · accepted · Accept

· Academic Editor

Accept

You have made excellent progress with this manuscript and it is now very acceptable for publication